# Recent Advancements in Microbial Polysaccharides: Synthesis and Applications

**DOI:** 10.3390/polym13234136

**Published:** 2021-11-26

**Authors:** Yehia A.-G. Mahmoud, Mehrez E. El-Naggar, Ahmed Abdel-Megeed, Mohamed El-Newehy

**Affiliations:** 1Department of Botany and Microbiology, Faculty of Science, Tanta University, Tanta 31527, Egypt; yehia.mahmoud@science.tanta.edu.eg; 2Textile Research Division, National Research Center (Affiliation ID: 60014618), Cairo 12622, Egypt; 3Department of Plant Protection, Faculty of Agriculture Saba Basha, Alexandria University, Alexandria 21531, Egypt; hekemdar@yahoo.com; 4Department of Chemistry, College of Science, King Saud University, Riyadh 11451, Saudi Arabia; 5Department of Chemistry, Faculty of Science, Tanta University, Tanta 31527, Egypt

**Keywords:** exopolysaccharides, biopolymers, food industry, microbial polysaccharides

## Abstract

Polysaccharide materials are widely applied in different applications including food, food packaging, drug delivery, tissue engineering, wound dressing, wastewater treatment, and bioremediation sectors. They were used in these domains due to their efficient, cost-effective, non-toxicity, biocompatibility, and biodegradability. As is known, polysaccharides can be synthesized by different simple, facile, and effective methods. Of these polysaccharides are cellulose, Arabic gum, sodium alginate, chitosan, chitin, curdlan, dextran, pectin, xanthan, pullulan, and so on. In this current article review, we focused on discussing the synthesis and potential applications of microbial polysaccharides. The biosynthesis of polysaccharides from microbial sources has been considered. Moreover, the utilization of molecular biology tools to modify the structure of polysaccharides has been covered. Such polysaccharides provide potential characteristics to transfer toxic compounds and decrease their resilience to the soil. Genetically modified microorganisms not only improve yield of polysaccharides, but also allow economically efficient production. With the rapid advancement of science and medicine, biosynthesis of polysaccharides research has become increasingly important. Synthetic biology approaches can play a critical role in developing polysaccharides in simple and facile ways. In addition, potential applications of microbial polysaccharides in different fields with a particular focus on food applications have been assessed.

## 1. Introduction

Polysaccharides are the main constituent of carbohydrates in nature are consist of monosaccharides as building blocks [1,2]. There are two forms of polysaccharides; homopolysaccharides, which are comprised of only one type of monosaccharide molecule and heteropolysaccharides, which contain more than one type of monosaccharides [3]. D-glucose is the most predominant component of polysaccharides [4]. Additionally, there are other types of monosaccharides derivatives occurring in polysaccharides, such as simple sugar acids (glucuronic and iduronic acid), amino sugars (D-galactosamine and D-glucosamine), and amino sugars’ derivatives (N-acetylneuraminic acid and N-acetylmuramic acid) [5]. Homopolysaccharides are termed according to the unit of sugar they contain; hence, glucans refer to glucose homopolysaccharides, while mannans refers to mannose homopolysaccharides.

Several untraditional polysaccharides can be identified from photosynthetic plants, algae, parasites, and microorganisms. Various polysaccharides have antitumor, antimicrobial, immune-regulating, and hypoglycemic properties [6]. Natural polysaccharides can boost the immune system and assist with a variety of diseases. Moreover, polysaccharides are widely applied in the food and dairy industries [5,7].

The utilization of polysaccharides as a carrier for anticancer drugs has attracted much attention over the past decade [8,9]. Polysaccharides from macro-fungi might be utilized as a multi-purpose prescription in the future decade [10]. Furthermore, with the rapid advancement of science and medicine, plant polysaccharide research has become increasingly important [8,11]. Despite the fact that there are many unexplained mysteries in this regard, plant polysaccharides have therapeutic effects and a wide range of applications, making them an essential route for developing novel drugs. Medications and antimicrobials derived from various plant polysaccharides have been utilized to improve sedative vitality or lessen symptoms [12,13]. The immunological regulation mechanism of bioactive plant polysaccharides is becoming clearer as research progresses, and more bioactive plant polysaccharides would be developed and employed in different applications [6].

There are also few famous instances (dextran and levans) in which the development and polymerization occur outside cells due to naturally occurring enzymes that transform the substrate into the polymer in the extracellular environment [14,15]. The heterogeneity of polysaccharides is attributable to the wide variety of compositions of monosaccharides and linkages. Consequently, no ubiquitous digestive enzyme exists for generating oligosaccharides. With their strong affinity for particular glycosidic bonds, natural glycosidase enzymes are noteworthy. Thus, a particular enzyme could only break a limited number of polysaccharide linkages [16,17]. Polysaccharides are not only an energy source and supportive tissue structure, but they are more significantly active in biochemical processes, including viral and bacterial infection, cell proliferation, biosynthesis, intestinal flora, and metastasis of tumor cells [18]. Polysaccharides are distinguished based on their monosaccharide constituents, their chain length, and the number of branches [19]. In several sectors, the polysaccharide family generated from microbes has an essential role in the food processing industry [20]. Several factors such as temperature, pH, growing media components, and the availability of oxygen can influence microbial polysaccharides and play a significant role in the physiological processes of microorganisms [21,22,23].

Kim et al., assessed the effects of agitation strength and aeration rate on developing antioxidant exopolysaccharides from the submerged *Ganoderma resencium* mycelial community [24]. They inferred that the factors of agriculture seemed to have a remarkable impact on the structure of exopolysaccharide (EPS) carbohydrates, resulting in various antioxidative behaviors [25]. It implied that processed food packages offered in the market nowadays had heightened food consistency standards from ready-to-eat meals. In addition, other substantial polymers contribute to the polysaccharides present in living organisms called biopolymers [26]. Biopolymers have been more common than conventional polymers in recent decades, even though they are natural and environmentally sustainable [27]. Natural biopolymers, mainly polysaccharides that are generated from trees, plants, tree gums, seeds, tubers, seaweed, are the most utilized carbohydrate polymers in many different applications [28].

Herein, this review article focuses on the details of the bacterial biosynthetic pathways for polysaccharides. In addition, we highlighted the targeted polysaccharides structures modifications using molecular biology tools. The review article was extended to demonstrate microbial polysaccharides and their cellular functions. Some of the polysaccharide’s applications in food processing industries are also demonstrated, such as cellulose, xanthan, dextran, pullulan, glucans, chitin, and chitosan. Finally, the various applications of microbial polysaccharides in medical, health, wastewater, and bioremediations have also been reviewed.

## 2. Biosynthetic Pathways

Biopolymers are polymers that are produced during all organisms’ growth cycles under natural conditions. Consequently, they are also known as natural polymers such as carbohydrates [29,30]. Biopolymers are produced through complex metabolic processes within the microorganism’s cells [31]. Examples of biopolymers accumulated in the cytoplasm of cells are polyphosphate, polyhydroxyalkanoates, starch, cyanophycin, and glycogen [32]. Biopolymers are also isolated and identified from algae that thrive in natural conditions and some other rare natural sources [32]. It could also be produced in vitro in a cell-free system with isolated enzymes [32]. In addition, biopolymers are generated by the fermentation process. Intracellularly or extracellularly, the biotechnological production of biopolymers can proceed [33,34].

Figure 1 represents the synthesis pathway of microbial EPS. The bacterial biosynthetic pathways include (a) substrate digestion [35], (b) main metabolite pathway [36] and (c) polysaccharides synthesis [37,38]. Depending on the form of the substrate, either passive or active transport system, it may be absorbed by the cell, and catabolized through intracellular phosphorylation, or perhaps a direct oxidative periplasm could pass and oxidize it [39]. The periplasmic oxidative route only exists in some of these bacteria, although the intracellular phosphorylative pathway is widespread for bacteria. Both of these systems can be developed and work effectively in many EPS-producing species if substrate availability is offered [40].

Glycolysis is used to catabolize the substance in the cytoplasm. The production of polysaccharides entail the biosynthesis of effective precursors, which are energy-rich monosaccharides that can be generated from phosphorylated sugars, predominantly nucleoside diphosphate sugars (NDP-sugars) [42]. EPS secretion is a complicated bacterial mechanism wherein the hydrophilic polymers of high molecular weight packed in the cytoplasm get to cross the cell membrane without breaching the critical features of the barrier [43].

In most gram-negative bacteria, where a wide range of molecular structures is presented in EPSs, one of two mechanisms has been identified to follow their biosynthesis and release [41]. The main components of this integration system are the Wzx flippase, Wzy polymerase, and Wzz chain-length regulator proteins of the integral inner membrane that have defied detailed structural and functional characterization [44]. The polymer repeat complex is formed and polymerized on the periplasm, mostly on the inner surface of the cytoplasmic membrane and the transporter-dependent phase of ABC wherein the cytoplasmic surface of the inner membrane is polymerized [45].

Such polysaccharides provide potential characteristics to transfer toxic compounds and decrease their resilience to the soil [46,47]. New approaches, such as using modified genetically engineered microorganisms, were used to generate polysaccharides to increase the yield and allow economically efficient production [48,49].

The metabolic network prediction may also enable the understanding of relevant factors for rapid multiplication. Mannitol was established from the genome sequence mostly as stimulator for *Chromohalobacter*
*salexigens’* medically essential levan (a fructose homopolymer) biosynthesis [48]; a positive result was later obtained in another halophilic bacterium, *Halomonas smyrnensisis* [50]. Approaches of metabolic engineering could support the continuing fermentation process engineering efforts to optimize the development of EPS [51]. In this regard, as examined by Wang et al. [52] study on capsular K5 polysaccharides of *Escherichia coli* (K5 EPS) biosynthesis is a great illustration.

The fermentation method for heparosan has been optimized for improved production and yield. Transformation of uridine diphosphate glucose (UDP glucose) into UDP glucuronic acid via UDP glucose dehydrogenase and UDP-*N*-Acetyl-glucosamine pathway was established as controlling steps to maintain a balanced heparosan biosynthesis and cell wall synthesis supply of nucleotide sugars [53]. The requirement of a controlled over-expression of KfiA and KfiC GTs was discovered by genetic engineering targeting these metabolic reactions. The K5 lyase gene accomplished the breaking of such association has also been genetically engineered to increase the amount of K5 released in the supernatant, as part of the K5 polysaccharide, which directly binds to the cell membrane [52]. To regulate the length of chain, the K5 lyase gene could also be genetically engineered [52].

## 3. Targeted Polysaccharides Structures Modifications with Molecular Biology Tools

Carbohydrates as biomolecules can be used to address the resistance to medications that degrade in the battle of bacterial infections [54]. Recently, the interaction between carbohydrates and lectin has been widely studied [55]. However, their capacitance in many other processes, such as metabolism, is still not well established and requires more interdisciplinary joint investigations. Carbohydrates are inadequate for biological investigations due to their structural complexity and production difficulties [56]. Xu and Liu [57] proposed a top-down approach to peptidoglycan oligomers, which has offered a fresh perspective throughout this area. However, more common techniques, particularly possible automatic synthesis, are still needed. Furthermore, it is a critical emergency, and further development of glycobiology will encompass the biological route of the majority of sugar molecules at a molecular level [58].

Rehm [59] addressed strategies to manufacture modified biopolymers with modified characteristics of the metal, which increase the performance of the application. For the in vivo production of polymers, developed strains with enhanced yields and/or that really can generate modified or tailor-made polymers have been established. This analysis refers to metabolic engineering, applying the knowledge of pathways of polymer biosynthesis, metabolic flux, and main enzymes to alter the pathway of biosynthesis [60,61] correctly. There are two fundamental techniques, which can be in vitro implemented [60,61].

Firstly, the in vitro synthesis can lead to new biopolymers by using polymerizing or engineered enzymes subjected to specified substrates [59]. Secondly, exposure to enzymatic or chemical modifications may improve the isolated biopolymers [62,63], as shown in Figure 2.

This would be clearer if, along with biosynthetic genetic clusters, further EPS’ structure is identified with structure/bioactivity relationships. Reconstruction of the EPS biosynthesis pathway becomes conceivable with genomic sequences and functional notes and hence can be designed to acquire tailored polymers [64]. Unless the mechanisms of carbohydrate biosynthesis are established, the over-expression or the inhibition of targeted genes can lead to better development of EPS. The molecular weight, functional substituents, and acidic structure may also allow better vision into protein function and even the biosynthesis of tailored molecules with desirable features [65]. Protein engineering has adjusted the behavior of Glycosyltransferases—Gum genes (GT GumK) needed for xanthan synthesis) implicated in xanthan biosynthesis, resulting in a shift in the development of xanthan yield, emphasizing the opportunity to obtain tailor-made xanthan molecules thru GT engineering [66].

In *Sphingomonas* sp., examining the genes involved in diutan’s molecular weight control (repeated configuration of rhamnose, glucose, glucuronic acid, glucose units) were not effective [67]. This could imply that chain length regulation is very complicated regardless of the proposed biosynthesis mechanism [68].

GDP-mannose dehydrogenase has been widely established in *Pseudomonas aeruginosa* as a major regulatory protein in alginate biosynthesis via over-expression experiments [69]. On the other hand, genomic data will also enable the identification of new glycopolymer modifying enzymatic tools. Therefore, in the case of in vitro synthesis or in vivo biosynthesis engineering, enzymes would be interesting biotechnological tools. Marine biodiversity has also shown tremendous potential as biocatalysts [70]. Delbarre-Ladrat et al., have examined the exopolysaccharides generated by marine bacteria, including their implementations through glycosaminoglycan-like molecules [53,71,72].

Depolymerizing enzymes are being utilized in the process of in vitro depolymerization and as instruments for studying the chemical composition [73,74]. During in vitro synthesis, carbohydrate sulfotransferases and other replacement grafting enzymes, including acetate kinase, could be used to attach or remove substituents that are of critical interest to the molecule’s final bioactivity [75].

The classification of GTs is hard to categorize, particularly if they belong to a polyspecific family 4 or 2. Recognizing that both the chemical structure of the polysaccharide and the genetic cluster for biosynthesis will provide details for genetic knockout to explain the GT enzymatic role in the biosynthetic pathway [76,77]. It will also provide valuable guidance for enzymatic specificity determination and enzyme characterization. Whenever it was established that the polysaccharide biosynthesis cluster comprises cloning the entire biosynthesis cluster into a suitable heterologous host, recombinant development can be envisaged [78]. This is a modern approach established in accordance with the growth of synthetic biology [79,80,81]. Necessary to refer, only fewer complex polymers, including hyaluronic acid [82,83], chondroitin [84], and heparosananic acid [85], have indeed been identified in recombinant development.

Scientists have established groundbreaking methods throughout modern biotechnology to investigate and control living systems. Genetic engineering enables the time, location, degree, and form of gene expression to be controlled [86]. For protein polymers, the simplest case was applied. Accessing a specific protein polymer’s genetic blueprint (gene) enables one to alter both the polymer generating mechanism and the polymer’s structure. As a matter of fact, recombinant DNA techniques are considered to enable the development of identical (in length, composition, and spatial orientation) polymer chains [87].

### 3.1. Role of Genetics in Polymers Production Systems

Genetically engineered products are regulated based on their expected use, regardless of their manufacturing method or process [88]. For instance, genetically engineered foods are treated similarly to common ones under prevailing Food and Drug Administration (FDA) regulations. FDA does not require approval or labelling of new products, given that such products are identical in composition, function, and structure to the traditional food products. The genetics of the biosynthesis of the EPS relies on the microorganism involved and the generated polysaccharide [89]. That being said, certain general processes may be defined, focusing on the sort of polysaccharide [59]. Over the former decade, the molecular mechanisms and regulatory processes influencing the synthesis of biopolymers have been well established.

This available knowledge about the effective tools for engineering bacteria that can produce effective biopolymers and even synthetic polymers with specific material properties for particular high-value requirements, all at a feasible cost. Valuation relies on both the characteristics of the material and the relatively inexpensive of its processing [59].

### 3.2. Role of Genetics in Polysaccharides Over-Expression

For the application of genetic engineering, several mentioned techniques could be used to produce new and tailor-made molecules [90].

#### 3.2.1. Recombinant of Hyaluronic Acid (HA) and Alginate Production

The engineering and production of HA has been carried out using the metabolic engineering of bacterial strains and pathogenic natural producer as described by Cimini et al. [83]. Novozymes for HA biosynthesis have used the gram-positive bacteria (*Bacillus subtilis*), in several industrial processes. Additionally, to HA synthase, it retains all catalyst activities essential for the synthesis of hour angle [91].

Recombinant development of alginate GDP-mannose dehydrogenase, but at the other hand, catalyzes the irreversible process that changes GDP-mannose into GDP-mannonate through tightly controlling alginate synthesis [92,93].

#### 3.2.2. Biopolymer Production: Intracellular vs. Extracellular 

The existence of a space in the cytoplasm restricts the generated quantity of polymer by the cell [94], although most microorganisms are used for fermentative manufacturing processes. They also claimed that the density of the cell and the biopolymer fraction in the biomass are limited/determined by the yield per volume. Poly(-Dglutamate), as well as several polysaccharides including xanthan, dextran, alginates, curdlan, chitosan, pullulan, and microbial cellulose, are examples of a biopolymer that appears outside of the cells as a response for the cell excretion or extracellular production [95]. Both of these methods have already come true, and several different scale instances have been seen both in the laboratory and on a broad scale. For example, polylactic acid has been developed on a large scale through combination of biotechnological and chemical approaches [96]. Additionally, both methods have previously proven effectiveness, and several scaled examples have been developed on both a laboratory and big scale. For instance, polylactic acid has been produced on a large scale via a combination of chemical and biotechnological approaches [96]. Table 1 summarizes most of the microbial polysaccharides and their applications for genetic engineering purposes.

## 4. Cell Functions of Microbial Polysaccharides

There are multiple functions for microbial polysaccharides, notably exopolysaccharides (EPS). There is an increasing research interest in EPS from both prokaryotes and eukaryotes. The EPS generated by microorganisms is of significant concern because of its chemical, structural versatility as well as its physical, rheological and other promising applications in the food, bioremediation, wastewater treatment, bioleaching, biomedical and pharmaceutical fields [99,100]. Most of the synthesized polymers have a good degree of stability and degradation resistance; these would be deposited in the atmosphere, as described by some researchers, at such a rate of almost 8% by weight and 20% by landfill volume. Polymers are a category of “giant” molecules made up of different key components that are joined together, forming long chains [101].

Monomers, as simple building blocks, are the repeated units and often pointed to as “more complicated building blocks”. That being said, synthetic polymers have been cited as a significant concern, including plastics’ waste and water-soluble in wastewater. An active aspect of our everyday life is polymers-based plastics. Biopolymers, including cellulose and starch, are by far the most promising materials for biomedical applications. Fortunately, more complex hydrocarbon polymers are produced by bacteria and fungi, polysaccharides such as curdlan [102,103,104,105], xanthan gum, chitin, pullulan [106,107], chitosan, and hyaluronic acid, are becoming increasingly essential.

## 5. Applications of Different Polysaccharides

Several polysaccharides have been widely applied in various applications. The utilization of polysaccharide biopolymer materials for drug delivery may decrease the reaction of tissues and enable the non-conventional administration of drugs. Until now, the use of biopolymers in these compositions has also been limited to a restricted range of applications. The formulation of drug implementing systems for the delivery of polymer drugs is expected to influence the pharmaceutical industry shortly [108,109].

The treatment of cancers and diseases is a key area of application for these innovative sustained-release systems. Biopolymer delivery capabilities require drugs to be prescribed periodically when treating hormone disorders and geriatric diseases [110,111].

Dextran is thought to become the first polysaccharide generated by lactic acid bacteria (LAB) for commercial use. Many dextrans are soluble in water, while others aren’t. Dextran can be utilized to increase viscosity, water absorption, and sugar crystallization prevention.

In the production of ice cream, it acts as a gelling agent. Furthermore, owing to its great features, including such biocompatibility, biodegradability, adsorption, and the ability to create films for chelating metal ions, chitosan is gaining increasing economic attention as a useful resource material [94].

### 5.1. Food Applications of Different Polysaccharides

Table 2 lists the different utilized polysaccharides in foods applications such as cellulose, xanthan, dextran, pullulan, glucans, chitin and chitosan. 

#### 5.1.1. Cellulose

The most abundant biomass component and the primary feedstock for the paper and pulp industries is cellulose [103,137,138,139,140,141,142,143,144]. It can be extracted from plant tissues such as trees, cotton, etc. In addition, it can be generated by bacterial fermentation, resulting in a purer source of cellulose with distinct properties. The most typical bacterial cellulose (BC) products are attracted owing to its large surface area and capacity to withstand liquids [137,145]. As a result, very low amounts of BC can create excellent binding, thickening, and coating agents. It is expected to have many implementations in food due to its thickening properties, as seen in Table 1. For example, cellulose can be utilized as a thickening agent in soaps and a texture modifier in dough [112]. Moreover, cellulose was used as a shortening replacer in dough products, probably due to its thermo-gelling ability [146].

#### 5.1.2. Xanthan

A bacterium-produced xanthan is one of the first economically viable bacterial polysaccharides utilized for production [147]. Throughout the xanthan polymer building blocks or “repeat units”, the bacterium Xanthomonas forms five distinct sugar groups [148,149]. Thanks to its excellent gelling agents, Xanthan is being used in several food applications such as cheese spreads, ice creams, puddings, and other deserts [150,151]. According to Zia et al. [152], one of the key priorities of milk farmers is to produce dairy products of high quality. Xanthan could help to improve the texture and thus improve the quality of milk products.

#### 5.1.3. Dextran

As a natural polysaccharide, Dextran is mainly composed of monomers (simple sugars) and is maintained by yeast and bacteria. Dextran refers to a group of microbial polysaccharides that are formed or polymerized outside of the cell using an enzyme known as dextran sucrases [153,154,155]. Dextran is being applied in several food applications, including the improvement of characteristics of dough products [137,153]. Also, it is applied for enhancing the viscoelastic properties of frozen foods [130,131,132,156]. In addition, dextran polymers have several medicinal uses, such as wound dressings, surgical sutures, blood volume expanders boost blood supply in capillaries in the treatment of artery occlusion, and combat iron-deficient anemia. Dextran haemoglobin molecules are employed as blood supplements as well as plasma expanders for oxygen delivery qualities [157].

#### 5.1.4. Pullulan

As is known, pullulan is a water-soluble polysaccharide formed by many yeast species, most prominently, *Aureobasidium pullulans*, outside the cell. Pullulan is a linear polysaccharide composed of monomers joined together through three glucose sugars [106,107]. In biologically active conditions, pullulan compounds are biodegradable polymers. It is heat tolerant and possesses a wide variety of elasticities and solubilities. They can be found in several forms because of their versatility. This could be used as a bulking agent and texture enhancer in foods. It has no flavor, odor, or toxicity. It has a little caloric content, so it does not break down the components of naturally produced digestive enzymes [158].

Consequently, instead of starch or other fillers, it is being used as a food additive in low-calorie foods and beverages. Strong fertilizers use pullulan as a binding agent. Potash clays, uranium clays, and ferric hydroxide may be precipitated through slurries used throughout mineral or beneficiation with the biopolymer as a natural coagulant agent. Pullulan is also used in medicine as a plasma expander with no serious side effects as it is fully defecated after metabolic turnover. Pullulan compounds are being used as both drug carriers and surgical adhesives. Pullulan seems to have long-term commercial potential, because it is in the early stages of development [158].

### 5.2. Other Applications of Polysaccharides

#### 5.2.1. Health Aspects

Some LAB strains are becoming increasingly utilized as probiotics. Apart from acid and bile resistance, a probiotic LAB strain must be able to develop antimicrobial compounds towards pathogenic and cariogenic bacteria, as well as bind to and colonize human intestinal mucosa [159]. The development of antimicrobial compounds can help probiotics colonize the gut mucosa by giving them a competitive advantage over typical gastrointestinal microflora. Capsular polysaccharide has been shown to facilitate bacterial adhesion to biological surfaces, making it possible for bacteria to colonize different ecological niches [160].

The value and prospective applications of *Lactobacillus* plantarum, *Azotobacter vinelandii*, and associated with polysaccharides in fresh and sea water fish farms have been investigated [161]. The utilization of polysaccharides for probiotic bacteria microencapsulation increased the survival rate of encapsulated probiotics [162,163]. Most polysaccharide features, such as enteric dissolution, ion-induced gelation, and electrostatic activity, are ideal for forming microcapsules. Polysaccharides could also protect the encapsulated probiotic bacteria from the harsh environment [164,165,166]. A semipermeable membrane forms as the negatively charged property of alginate interacts with the positively charged property of chitosan, providing the corresponding capsules as strong texture and lower permeability to water-soluble molecules [167]. Importantly, by transferring the chitosan-coated beads into such an alginate/chitosan solution, the single-layer chitosan-coated alginate beads may be coated often, hence increasing the viability of the probiotic bacteria. (Figure 3).

Furthermore, according to Chen et al. [168], The microencapsulation of probiotic bacteria could enhance the quality, storage, and viability of delivered bacteria throughout the gastrointestinal tract. Polysaccharides are one of the most frequently used probiotic microencapsulation products. However, as the demand for probiotic microcapsules grows and new applications emerge, the use of polysaccharides and other widely used ingredients is facing many challenges. As a result, new forms of polysaccharides with primary properties appropriate for probiotic microencapsulation are required.

#### 5.2.2. Medical Application

The saccharide part of many vaccines is conjugated to a carrier protein, resulting in highly efficient and safe vaccines. Polysaccharides from marine *Vibrio* and *Pseudomonas* have been shown to have antiviral, antitumor, and immune-stimulant properties. *Alteromonas infernus*, a low molecular weight heparin-like EPS with anticoagulant effects, was discovered from deep-sea hydrothermal vents [169]. Clavan, and L-fucose, containing polysaccharides, can be utilized to prevent lung tumor cell colonization, control the formation of white blood cells, treat rheumatoid arthritis, synthesize antigens for antibody synthesis, and as a skin moisturizing agent in cosmeceuticals [170]. Because of their inherent abundance in plants, bioderived polymers including such polysaccharides have really been widely exploited in wound dressing production [171,172,173] (Figure 4).

Glucans are a homopolymer of glucose, a basic sugar joined together through β-(1,3) glycosidic bonds, with a connected side-chain of glucose residues linked together by β-(1,6) linkages. These glucans tend to have the ability to cure several diseases. β-glucans make up approximately half of the mass of the fungal cell wall [174]. Glucan component of the yeast cell wall is generally referred to as glucan. *Saccharomyces cerevisiae* is a typical source of this glucan, but it can also be presented in other areas. In yeast, glucans seem to be the most common polymers, accounting for 12 to 14% dry cell weight [175]. Glucan can be effectively refined from yeast cells by extracting all other cellular materials with a warm alkali solution, allowing the insoluble glucan material to recover. Several studies have been undertaken to examine the use of glucan as an anti-infectious agent due to its immunomodulatory properties. Plants are also using glucan as an antiviral agent. Glucans have been shown to suppress tumor development in mice and rats in many experiments using multiple tumor models. They also have the curious property of being radioprotective, which raises the odds of laboratory specimens surviving radioactive doses that would otherwise kill them [174].

Chitin and chitosan have a chemical structure that is comparable. Chitin is a linear polymer of acetylglucosamine groups, whereas chitosan is made by removing enough acetyl groups (CH_3_-CO) from the molecule to have it dissolve in diluted acids [176]. Despite having positive ionic charges, unlike plant fiber, chitosan could chemically bind to negatively charged lipids, metal ions, fats, proteins, cholesterol, and macromolecules. Chitosan is produced industrially by chemically deacetylating chitin. This approach does not produce high-quality chitosan. Deacetylation removes acetyl groups from chitin’s molecular chain, leaving chitosan with a chemically reactive amino group [94]. Chitin is a type of skeletal polysaccharide found in the shells of crabs, lobsters, shrimps, and insects. Chitin can be derived from many sources, such as shellfish and crustacean waste materials. Chitin is water-insoluble in its natural state, but chitosan, a partially deacetylated form of chitin. Because of their water-retention and moisturizing qualities, they are commonly used in the cosmetics industry. Chitin and chitosan are being used as carriers in the production of water-soluble drugs [177,178].

#### 5.2.3. Emulsifiers

Surfactants and emulsifiers extracted from bacteria are gaining much attention due to their biodegradability and ability to be manufactured from renewable resources [179]. As for emulsion, it has been found that EPS formed by a marine bacterium with various hydrocarbons can produce stable emulsions [180].

Exopolysaccharide (EPS 71a) formed by a marine *Enterobacter cloaceae* emulsified benzene, hexane, kerosene, xylene, and oils of coconut, paraffin, cottonseed, jojoba, groundnut, castor, and sunflower, according to Ask et al. [181].

At an optimum concentration of 1 mg/mL with peanut oil and hexane, it might generate emulsions. Increasing the concentration of EPS 71a did not result in a substantial improvement in the properties of emulsions.

As previously known, emulsions are being used to give specialized physicochemical qualities and functional features in pharmaceutical, food, and personal care sectors. In many cases, it is preferable to build emulsions with natural components in order to generate “label-friendly” items. Corn fiber gum, Gum arabic, and beet pectin have been used. These three polysaccharide-based emulsifiers were capable of producing emulsions, with average diameter decreasing as even the concentration of emulsifier with increasing the homogenization pressure. Moreover, monodisperse soybean oil-in-water emulsions were generated using the microchannel emulsification technique, using polysaccharides as the only emulsifier. The influence of several polysaccharide such as gum Arabic, carboxymethyl cellulose, sodium alginate, and pectin, with different concentrations on droplet size, droplet size distribution, and emulsion stability were explored [182]. Emulsions stabilized using sodium alginate and Arabic gum were shown to be stable for more than 5 h. On the other hand, the emulsifying potential of a polysaccharide that isolated from Cordia abyssinica was examined at different concentrations. The homogenizer was used to make emulsions of vegetable oil containing up to 1% polysaccharide in phosphate buffer (pH 7.4).

Whey protein isolate nanofibrils can be employed as a new stabilizer in the pickering emulsion system to increase lipophilic bioactive compounds’ water solubility, stability, and bioavailability. To make a stable pickering emulsion, researchers employed conjugated linoleic acid and whey protein isolate nanofibrils [183]. Such results demonstrated that a pickering emulsion supported via whey protein isolate nanofibrils is preferable for carrying conjugated linoleic linoleic acid because it boosts conjugated linoleic acid solubility and has much more active uses in biology and food as well [183,184].

In addition, nanofibrils manufactured from whey protein isolate nanofibrils were used to maintain a Pickering emulsion with a high internal phase. Hydrothermal at 110 °C for 4 h produced whey protein isolates with well-ordered sheet structures and a high aspect ratio, which would be more effective than the existing approach of 85 °C for 5–25 h. The findings showed that hydrothermal preparation of whey protein isolate nanofibrils is an unique and efficient approach, and as such the nanofibrils may be employed to sustain high internal phase Pickering emulsions with variable rheology [183].

In the presence of sodium chloride (range of 5–50 mg/mL), emulsions of groundnut oil and hexane were stable for 10 days at 35–37 °C and pH 2–10. Compared to conventional gums like Arabic gum, tragacanth, karaya, and xanthan, EPS 71a can form stable emulsions with xylene. It was found that EPS was more effective than commercially available emulsifiers [185].

#### 5.2.4. Polysaccharide-Based Edible Films

The food industry has a big difficulty in the form of food waste. In this context, edible films and coatings have gotten a lot of interest because of their capacity to prevent food spoiling during handling, transportation, and storage. This has greatly aided in the extension of the shelf-life of food goods. Polysaccharides, among other polymers, have been studied in the recent decade for the development of edible films and coatings. In microbial food safety applications, such polymeric frameworks have shown tremendous potential. The addition of essential oils to polysaccharide matrices increased the functional qualities of edible films and coatings [186].

The qualities of binary mix films based on gelatin and polysaccharides with emulsifying activity: water soluble soy polysaccharides, methylcellulose, gum Arabic, and octenyl succinic anhydride modified starch were examined. The stretchiest film was made by combining gelatin and octenyl succinic anhydride modified starch in a 25/75 ratio. During storage, however, this film showed clear signs of recrystallization [187].

From the other hand, when using salted duck egg yolk, which is a traditional pickled egg product, weight loss and texture degradation occur during storage. To better maintain the texture of salted duck egg yolk, an edible covering relying on whey protein isolate nanofibers containing plasticizer agent (glycerol) and carvacrol as an antibacterial agent was created. When compared to other samples, the whey protein isolate nanofibers-carvacrol/glycerol edible coating showed stronger antibacterial activity, while the whey protein isolate nanofibers-carvacrol/glycerol film had smooth and continuous surfaces and improved transmission [188].

Weight loss and variations in the textural qualities of salted duck egg yolk with whey protein isolate nanofibers with carvacrol/glycerol coating were also assessed. Salted duck egg yolks with whey protein isolate nanofibers—carvacrol/glycerol coating showed less weight loss, according to the findings. The whey protein isolate nanofibers-carvacrol/glycerol coating on salted duck egg yolk greatly enhanced the textural qualities compared to the untreated samples. The egg yolks covered with the whey protein isolate nanofibers-carvacrol/glycerol coating showed the least amount of hardening (18.22%).

As a result, whey protein isolate nanofibers-based coatings might have a bright future in the food business. The effect of soy protein isolate preheating temperature on the physicochemical properties of soy protein isolate-oil emulsion films was also investigated. In the soy protein isolate-oil emulsion films, there was no noticeable change in protein molecular weight as the preheating temperature was increased, although the glass transition temperature improved [189].

#### 5.2.5. Wastewater and Bioremediation

A emerging topic of biotechnology is the use of EPS-producing microbes in the treatment of mining-related environmentally favorable industrial wastewater [190]. Biofilm-mediated bioremediation is a more efficient alternative to planktonic bacteria-mediated bioremediation because cells developing inside a biofilm can adapt to the surrounding environmental conditions [191,192]. Biofilms support the cells to enhance the mineralization processes by preserving optimum physiological and chemical conditions, localized solute concentrations, and redox potential [193]. Biofilm reactors are often used to treat hydrocarbons, heavy metals, and large amounts of dilute aqueous solutions, such as commercial and municipal wastes. [193]. EPS plays an essential function in removing hazardous metals from the environment due to their participation in flocculation and high affinity for the metal ions from solutions [194].

Sulfate-reducing bacteria (SRB) are a common form of bacteria contained in metal-contaminated wastewater. This bacterial group is successful in the anaerobic oxidation of many organic compounds and the removal of wastewater heavy metals [195,196]. Bacteria such as *enterobacter* and *pseudomonas* can breakdown or absorb hazardous heavy metals during bioremediation methods [197,198].

## 6. Conclusions

Modification for the encode enzymes that catalyze the reactions in the processes or through modifying signaling pathways that influence gene expression and thus enzyme function, provides the most promising prospects towards increasing pure and high-quality EPS production. In the near future, more interdisciplinary studies are recommended to widespread screening, study structural characterization, develop bioactivity tests, understand biosynthesis and metabolic engineering problems. Thus, synthetic biology approaches can play a critical role in developing polysaccharides in simple and facile ways.

## Figures and Tables

**Figure 1 polymers-13-04136-f001:**
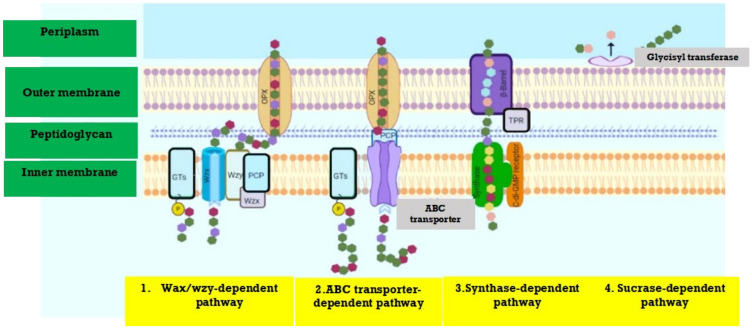
Synthesis pathway for producing EPS. TPR: tetratricopeptide repeat protein; OPX: outer membrane polysaccharide export; PCP: polysaccharide co-polymerase; GTs: glycosyltransferases, modified from [41].

**Figure 2 polymers-13-04136-f002:**
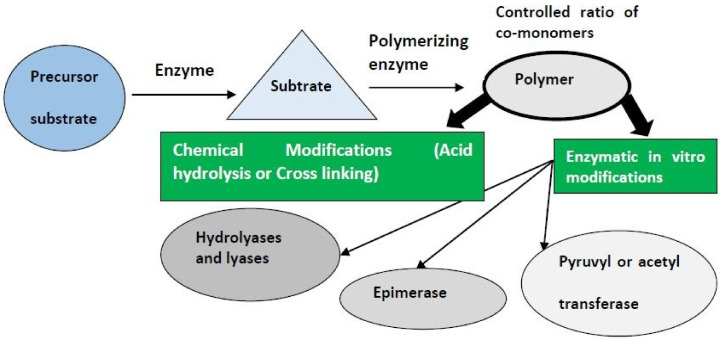
Proposed basic strategies for the in vitro modification of polysaccharides.

**Figure 3 polymers-13-04136-f003:**
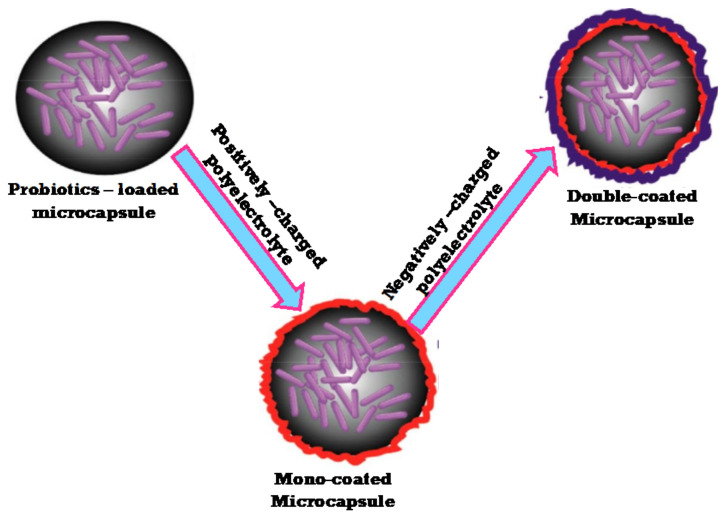
A diagram illustrates the alginate-chitosan beads with single- and double-layer coatings. Adapted from Ref. [167].

**Figure 4 polymers-13-04136-f004:**
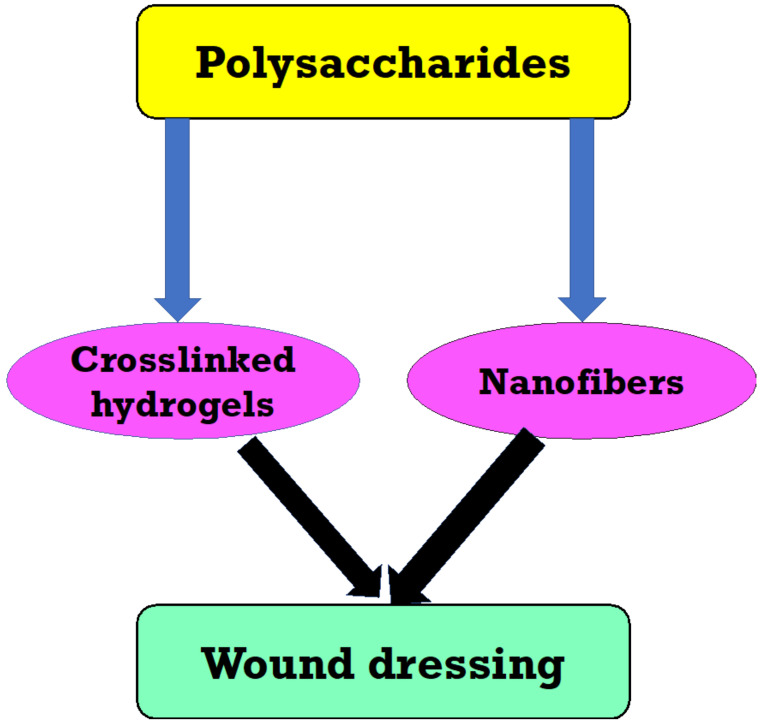
Fabrication and biocompatibility assessment of polysaccharide materials.

**Table 1 polymers-13-04136-t001:** Some of the genetic engineering applications in the production of microbial polysaccharides.

Genetic Engineering Applications	Microorganisms Involved	References
Recombinant Hyaluronan/hyaluronic acid (HA) production	*Bacillus subtilis* *E coli JM109*	[97] [98]
Recombinant Alginate and alginate production	*P. aeruginosa*	[69]
Intracellular versus extracellular production of biopolymers	-	[95,96]

**Table 2 polymers-13-04136-t002:** Polysaccharides in the food processing industry and their used concentration.

Polysaccharide	Types of Utilized Foods	Required Concentration (%)	Functionality	References
Cellulose	dough-based products		texture modifier	[112]
	creamed soups	0.75	Thickening agent	[112]
Xanthan gum	dressings for salads	0.1–0.5	Suspending agent, dispersant, and emulsion stabilizer	[113,114,115]
	Mixes that are dry	0.05–0.2	Disperses in hot or cold water with ease.	
	Sauces, toppings, relishes, and syrups	0.05–0.2	Heat retention and uniform viscosity are both qualities of a thickener.	
	Beverages	0.05–0.2	Stabilizing agent	
	Milk products	0.2–0.5	Stabilizing agent; controlling the viscosity control of mixture	
	Baked foods	0.1–0.4	Stabilizing agent	
	Frozen foods	0.05–0.2	Improves the stability of the freeze-thaw cycle	
Gellan	Jellies Jams Confectionery Processed foods Processed meats Icings Pie fillings	0.15–0.2 0.12–0.3 0.8–1 0.2–0.3 0.1–1 0.05–0.12 0.25–0.35	Agent for gelling Spreads with few calories Fruit and vegetable gelling Modification of texture Agent for coating Texturizer	[99,116,117,118,119]
Pullulan	Edible Films in confectionary decoration Snack Foods	5–10	Low oxygen permeability edible films, bioadhesive stability at high pH, and low viscosity NaCl	[120,121]
Dextran	Bakery products	2	Distinctive dough-mixing characteristics	[122,123]
	Ice cream, Frozen and dried foods	2–4	Beneficial viscosity properties Film of dextran used in frozen foods	
Xylinan/Acetobacterxylinum cellulose	Confectionery product-Nata de coco		Agent for gelling and controlling the viscosity	[113,124]
Alginates	Confectionery, Dairy products Beverages, Jams, Soups, Sauces, Meat, Fish, and gellies	0.3	Gelling agent, thickener, and stabilizer	[125,126,127,128,129,130,131,132]
Curdlan	Gellies	1–5	Agent for gelling	[102,103,133,134,135,136,137]
	Processed meats	1–10	Agent for gelling	
	Processed meats	0.1–1	Modification of texture	
	Sauces	0.2–0.7	Improving the mixture viscosity	
	Freeze-dried foods	0.5–1	Improving the mixture rehydration	

## Data Availability

The data that support the findings of this study are available on request from the corresponding author.

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
