# Peer review of "Recent Advancements in Microbial Polysaccharides: Synthesis and Applications"

_polymers, 2021, doi:10.3390/polym13234136_

Round 1

Reviewer 1 Report

In the manuscript “Microbial Polysaccharides for food processing industry and other purposes”, the author summarized the generation and numerous applications of microbial polysaccharides. This is a nice organized review article. The review works done by the authors were informative and valuable. The manuscript might be considered for publication after minor revision.

1. Edible film was one of the important applications of polysaccharides. However, it was not mentioned in the text. It was suggested to talk about the polysaccharide-based edible films based on the comparison with edible film of other natural materials. (doi:10.3390/foods9040449, doi:10.1016/j.jfoodeng.2021.110697)

2. There were too few references in Section emulsifier. The author should provide critical review on the polysaccharide emulsifier. It might be better to compare the innovative emulsifier based on polysaccharide and other natural materials, such as protein. (doi:10.3390/foods10081892, doi: 10.1016/j.foodhyd.2021.107180)

3. Figure 4 should be placed appropriately.

Author Response

In the manuscript “Microbial Polysaccharides for food processing industry and other purposes”, the author summarized the generation and numerous applications of microbial polysaccharides. This is a nice organized review article. The review works done by the authors were informative and valuable. The manuscript might be considered for publication after minor revision.

  1. Edible film was one of the important applications of polysaccharides. However, it was not mentioned in the text. It was suggested to talk about the polysaccharide-based edible films based on the comparison with edible film of other natural materials. (doi:10.3390/foods9040449, doi:10.1016/j.jfoodeng.2021.110697)

Thanks for your useful comments that actually enhanced our article review as follow:

The food industry has a big difficulty in the form of food waste. In this context, edible films and coatings have gotten a lot of interest because of their capacity to prevent food spoiling during handling, transportation, and storage. This has greatly aided in the extension of the shelf-life of food goods. Polysaccharides, among other polymers, have been studied in the recent decade for the development of edible films and coatings. In microbial food safety applications, such polymeric frameworks have shown tremendous potential. The addition of essential oils  to polysaccharide matrices increased the functional qualities of edible films and coatings [1].

The qualities of binary mix films based on gelatin and polysaccharides with emulsifying activity: water soluble soy polysaccharides, methylcellulose, gum Arabic,  and octenyl succinic anhydride modified starch were examined. The most stretchy film was made by combining gelatin and octenyl succinic anhydride modified starch in a 25/75 ratio. During storage, however, this film showed clear signs of recrystallization [2].

On the other hand, the utilization of salted duck egg yolk, which is one of the classic pickled egg products, suffers from weight loss and texture deterioration during storage. An edible coating based on whey protein isolate nanofibers with glycerol as a plasticizer and including carvacrol as an antibacterial agent was developed to better retain the texture of salted duck egg yolk. When compared to other samples, the whey protein isolate nanofibers – carvacrol/glycerol edible coating showed stronger antibacterial activity, while the whey protein isolate nanofibers –carvacrol/glycerol film had smooth and continuous surfaces and improved transmission [3]. Weight loss and variations in the textural qualities of salted duck egg yolk with whey protein isolate nanofibers With carvacrol/glycerol coating were also assessed. Salted duck egg yolks with whey protein isolate nanofibers–carvacrol/glycerol coating showed less weight loss, according to the findings. The whey protein isolate nanofibers – carvacrol/glycerol coating on salted duck egg yolk greatly enhanced the textural qualities compared to the untreated samples. The egg yolks covered with the whey protein isolate nanofibers – carvacrol/glycerol coating showed the least amount of hardening (18.22% ). As a result, whey protein isolate nanofibers-based coatings might have a bright future in the food business. Also explored was the influence of soy protein isolate preheating temperature on the physicochemical features of soy protein isolate - oil emulsion films.During increasing preheating temperature, there was no discernible change in protein molecular weight in the soy protein isolate - oil emulsion films, although the glass transition temperature improved [4].

  1. There were too few references in Section emulsifier. The author should provide critical review on the polysaccharide emulsifier. It might be better to compare the innovative emulsifier based on polysaccharide and other natural materials, such as protein. (doi:10.3390/foods10081892, doi: 10.1016/j.foodhyd.2021.107180)

Thanks for your useful comments that actually enhanced our article review as follow:

As previously known, emulsions are being used to give specialized physicochemical qualities and functional features in the food, pharmaceutical, and personal care sectors. In many cases, it is preferable to build emulsions with natural components in order to generate "label-friendly" items. Gum arabic, corn fiber gum, and beet pectin are all used. These three polysaccharide-based emulsifiers were capable of producing emulsions, with the mean particle diameter decreasing as even the emulsifier concentration and homogenization pressure increased. Moreover, monodisperse soybean oil-in-water emulsions were generated using the microchannel emulsification technique, using polysaccharides as the only emulsifier. The influence of several polysaccharide such as sodium alginate, carboxymethyl cellulose, pectin, gum arabic with different concentrations on droplet size, droplet size distribution, and emulsion stability were explored [5]. Emulsions stabilised using sodium alginate and gum arabic were shown to be stable for more than 5 h. On the other hand, the emulsifying potential of a polysaccharide that isolated from Cordia abyssinica was examined at different concentrations. The homogenizer was used to make emulsions of vegetable oil containing up to 1% polysaccharide in phosphate buffer (pH 7.4).

Whey protein isolate nanofibrils can be employed as a new stabiliser in the pickering emulsion system to increase lipophilic bioactive compounds' water solubility, stability, and bioavailability. To make a stable pickering emulsion, researchers employed conjugated linoleic acid and whey protein isolate nanofibrils [6]. Such results demonstrated that a pickering emulsion supported via whey protein isolate nanofibrils is preferable for carrying conjugated linoleic linoleic acid because it boosts conjugated linoleic acid solubility and has much more active uses in biology and food as well [6,7].

In addition, nanofibrils manufactured from whey protein isolate nanofibrils were used to maintain a Pickering emulsion with a high internal phase. Hydrothermal at 110 °C for 4 h produced whey protein isolate with well-ordered -sheet structures and a high aspect ratio, which would be more effective than the existing approach of 85 °C for 5-25 h. The findings showed that hydrothermal preparation of whey protein isolate nanofibrils is an unique and efficient approach, and as such the nanofibrils may be employed to sustain high internal phase Pickering emulsions with variable rheology [6].

  1. Figure 4 should be placed appropriately.

Figure 4 has been placed in its appropriate position

References

[1]      A. Anis, K. Pal, S.M. Al-Zahrani, Essential oil-containing polysaccharide-based edible films and coatings for food security applications, Polymers (Basel). 13 (2021) 575.

[2]      K. Łupina, D. Kowalczyk, E. Zięba, W. Kazimierczak, M. Mężyńska, M. Basiura-Cembala, A.E. Wiącek, Edible films made from blends of gelatin and polysaccharide-based emulsifiers - A comparative study, Food Hydrocoll. 96 (2019) 555–567. https://doi.org/https://doi.org/10.1016/j.foodhyd.2019.05.053.

[3]      Q. Wang, W. Liu, B. Tian, D. Li, C. Liu, B. Jiang, Z. Feng, Preparation and characterization of coating based on protein nanofibers and polyphenol and application for salted duck egg yolks, Foods. 9 (2020) 449.

[4]      Y. Hu, L. Shi, Z. Ren, G. Hao, J. Chen, W. Weng, Characterization of emulsion films prepared from soy protein isolate at different preheating temperatures, J. Food Eng. 309 (2021) 110697. https://doi.org/https://doi.org/10.1016/j.jfoodeng.2021.110697.

[5]      L. Bai, S. Huan, Z. Li, D.J. McClements, Comparison of emulsifying properties of food-grade polysaccharides in oil-in-water emulsions: Gum arabic, beet pectin, and corn fiber gum, Food Hydrocoll. 66 (2017) 144–153.

[6]      Y. Yang, Q. Jiao, L. Wang, Y. Zhang, B. Jiang, D. Li, Z. Feng, C. Liu, Preparation and evaluation of a novel high internal phase Pickering emulsion based on whey protein isolate nanofibrils derived by hydrothermal method, Food Hydrocoll. 123 (2022) 107180. https://doi.org/https://doi.org/10.1016/j.foodhyd.2021.107180.

Reviewer 2 Report

The review article "Recent Advancements in Microbial Polysaccharides: Synthesis and Applications" is very interesting and discussed the synthesis of microbial polysaccharides and their applications. Other than that they also discussed the polysaccharides applications in food industry. The article was written well and they have implemented all my previous suggestions in the revised manuscript. However, they have to improve the abstract of the manuscript. They have to provide more details in the abstract, which they discussed in the manuscript. Hence I recommend the article to publish after a minor revision. 

Author Response

The review article "Recent Advancements in Microbial Polysaccharides: Synthesis and Applications" is very interesting and discussed the synthesis of microbial polysaccharides and their applications. Other than that they also discussed the polysaccharides applications in food industry. The article was written well and they have implemented all my previous suggestions in the revised manuscript. However, they have to improve the abstract of the manuscript. They have to provide more details in the abstract, which they discussed in the manuscript. Hence I recommend the article to publish after a minor revision. 

Thanks for your positive comments, we tried to go over the introduced data in the text and summarize it in the revised abstract.

Abstract: Polysaccharide materials are widely applied in different applications including such food, food packaging, drug delivery, tissue engineering, wound dressing, wastewater treatment, and bioremediation sectors. They were used in these domains owing to their efficient, cost-effective, non-toxicity, biocompatibility and biodegradability. As known, polysaccharides can be synthesized by different simple, facile and effective methods. Of these polysaccharides are cellulose, sodium alginate, chitosan, chitin, curdlan, dextran, pectin, xanthan, pullulan and so on. In this current article review, we focused on discussing the synthesis and potential applications of microbial polysaccharides. The biosynthesis of polysaccharides from microbial sources has been considered. Moreover, the utilization of molecular biology tools to modify the structure of polysaccharides has been covered. Such polysaccharides provide potential characteristics to transfer toxic compounds and decrease their resilience to the soil. Genetically modified microorganisms not only improve yield of polysaccharides, but also allow economically efficient production. With the rapid advancement of science and medicine, biosynthesis of polysaccharides research has become increasingly important. Synthetic biology approaches can play a critical role in developing polysaccharides in simple and facile ways. In addition, potential applications of microbial polysaccharides in different fields with a particular focus on food applications have been assessed.

Reference

[7]      Q. Jiao, Z. Liu, B. Li, B. Tian, N. Zhang, C. Liu, Z. Feng, B. Jiang, Development of Antioxidant and Stable Conjugated Linoleic Acid Pickering Emulsion with Protein Nanofibers by Microwave-Assisted Self-Assembly, Foods. 10 (2021) 1892.

This manuscript is a resubmission of an earlier submission. The following is a list of the peer review reports and author responses from that submission.

Round 1

Reviewer 1 Report

In the manuscript “Microbial Polysaccharides for food processing industry and other purposes”, the author summarized the generation and numerous applications of microbial polysaccharides. This is a nice organized review article. The review works done by the authors were informative and valuable. The manuscript might be considered for publication in its present form.. There were still some issues, which could be done at the stage of proof.

Table 1 was not mentioned in the text.

Line 551: Table 1 should be changed to Table 2

Author Response

Reviewer #1

In the manuscript “Microbial Polysaccharides for food processing industry and other purposes”, the author summarized the generation and numerous applications of microbial polysaccharides. This is a nice organized review article. The review works done by the authors were informative and valuable. The manuscript might be considered for publication in its present form.

Response:

Thanks a lot for your positive comments regarding our review article.

There were still some issues, which could be done at the stage of proof.

Table 1 was not mentioned in the text.

Response:

Thanks for your comment, Table 1 has been cited in the revised manuscript.

Line 551: Table 1 should be changed to Table 2

Response:

The table has been correctly numbered.

Reviewer 2 Report

The review article "Microbial polysaccharides for food processing industry and other purposes" is very interesting and the article has been improved compared to the previous version. The Authors have been implemented my previous suggestions in the new version and I agree with the authors. However there are very few minor suggestions to improve the article, hence I recommend the article to publish after a minor revision.

Minor points:

Line 101-113: Please cite the reference articles appropriately at the end of the each sentence, Instead of giving all of them at the end of paragraph (19-21).

Line 160-167: Please cite a reference article.

Figure 1: Please increase the font size (bottom text) to see it more clearly visible.

Table 2: The citations in the table should be provided next to the appropriate functionality / concentration, instead of giving all the references together. 

Author Response

Reviewer #2

The review article "Microbial polysaccharides for food processing industry and other purposes" is very interesting and the article has been improved compared to the previous version. The Authors have been implemented my previous suggestions in the new version and I agree with the authors. However there are very few minor suggestions to improve the article, hence I recommend the article to publish after a minor revision.

Response:

Thank you for your positive comments and useful suggestions. We have made some revisions based on your comments to improve the quality of this manuscript.

Minor points:

Line 101-113: Please cite the reference articles appropriately at the end of the each sentence, Instead of giving all of them at the end of paragraph (19-21).

Response:

Thank you for your comment. The references have been cited as suggested by reviewer.

Line 160-167: Please cite a reference article.

Response:

We have checked the whole article review and most of its content has been cited.

Figure 1: Please increase the font size (bottom text) to see it more clearly visible.

Response:

Font size for Figure 1 has been increased to be more readable.

Table 2: The citations in the table should be provided next to the appropriate functionality / concentration, instead of giving all the references together. 

Response:

Each functional of the utilized polysaccharides has been separately cited as recommended by reviewer.

Reviewer 3 Report

The manuscript entitled "Microbial polysaccharides for the food processing industry and other purposes" by Mahmoud et al is scientifically not at all significant in the field of microbial polysaccharides and does not contain any new information. There are several reviews published in this field and the present article is nothing new. 

Author Response

Reviewer #3

The manuscript entitled "Microbial polysaccharides for the food processing industry and other purposes" by Mahmoud et al is scientifically not at all significant in the field of microbial polysaccharides and does not contain any new information. There are several reviews published in this field and the present article is nothing new. 

Response:

Our current review article is mainly focused on the definition of biopolymers, the bacterial biosynthetic pathways for polysaccharides. It also highlighted the targeted polysaccharides structures modifications with molecular biology tools. Additionally, the submitted article review aimed to demonstrate the microbial polysaccharides and their cellular functions. The review article was also mentioned in details some of the utilized polysaccharides that were used in food processing industries. Finally, the various applications of microbial polysaccharides in medical, health, waste-water and bioremediations have also been mentioned. Based on all these mentioned points, we considered that the submitted review article is original and has many of the points that completely different from the other published articles.

Round 2

Reviewer 3 Report

The manuscript does not add any new and significant information to the field of microbial polysaccharides, the arrangement and hypothesis of the review are not scientifically sound. Title, abstract, and conclusion are not related to the constituents of the manuscript, along with poor representation.